# CRISPR/Cas9 Directed Reprogramming of iPSC for Accelerated Motor Neuron Differentiation Leads to Dysregulation of Neuronal Fate Patterning and Function

**DOI:** 10.3390/ijms242216161

**Published:** 2023-11-10

**Authors:** Katie Davis-Anderson, Sofiya Micheva-Viteva, Emilia Solomon, Blake Hovde, Elisa Cirigliano, Jennifer Harris, Scott Twary, Rashi Iyer

**Affiliations:** 1Bioscience Division, Los Alamos National Laboratory, Los Alamos, NM 87544, USA; kdavisanderson@lanl.gov (K.D.-A.); esolomon@lanl.gov (E.S.);; 2Department of Psychology, University of British Columbia, Vancouver, BC V6T 1Z4, Canada; 3Information Systems and Modeling Division, Los Alamos National Laboratory, Los Alamos, NM 87544, USA; 4Physical Chemistry and Applied Spectroscopy, Los Alamos National Laboratory, Los Alamos, NM 87544, USA

**Keywords:** CRISPR/Cas9 gene editing, iPSC reprogramming, motor neuron development, global transcriptome, electrophysiological activity

## Abstract

Neurodegeneration causes a significant disease burden and there are few therapeutic interventions available for reversing or slowing the disease progression. Induced pluripotent stem cells (iPSCs) hold significant potential since they are sourced from adult tissue and have the capacity to be differentiated into numerous cell lineages, including motor neurons. This differentiation process traditionally relies on cell lineage patterning factors to be supplied in the differentiation media. Genetic engineering of iPSC with the introduction of recombinant master regulators of motor neuron (MN) differentiation has the potential to shorten and streamline cell developmental programs. We have established stable iPSC cell lines with transient induction of exogenous *LHX3* and *ISL1* from the Tet-activator regulatory region and have demonstrated that induction of the transgenes is not sufficient for the development of mature MNs in the absence of neuron patterning factors. Comparative global transcriptome analysis of MN development from native and Lhx-ISL1 modified iPSC cultures demonstrated that the genetic manipulation helped to streamline the neuronal patterning process. However, leaky gene expression of the exogenous MN master regulators in iPSC resulted in the premature activation of genetic pathways characteristic of the mature MN function. Dysregulation of metabolic and regulatory pathways within the developmental process affected the MN electrophysiological responses.

## 1. Introduction

As the world population ages, neurodegenerative diseases are increasing. The outcome of these diagnoses varies but can lead to fatality 50% of the time within 15–20 months [1]. Treatments for neurodegeneration, in particular motor neuron (MN) diseases, are restricted due to the limitations of motor neuron repair and regeneration. Induced pluripotent stem cells (iPSCs) have been proposed as a therapeutic agent due to their ability to successfully differentiate into motor neurons [2,3]. However, standard protocols for iPSC differentiation into motor neurons are laborious and can result in mixed populations of motor neuron subtypes [2,3,4,5]. Therefore, further optimization is needed to leverage iPSC’s therapeutic potential.

iPSC can be programmed into MN lineages using media supplemented with growth factors, pluripotency inhibitors, and other neuron patterning factors [2]. Alternatively, iPSCs can be genetically reprogrammed to express master regulators of MN development for controlled and rapid progress via neuronal differentiation by reducing or eliminating the intermediate progenitor states [6,7]. It has been reported that the expression of three key transcription factors, i.e., NGN2, ISL1, and LHX3, in mouse embryonic stem cells (mESC) is sufficient for the establishment of motor neuron identity [6,7,8]. Neurogenin 2 (NGN2) is normally expressed in neuronal progenitor cells (NPC) to commence neural differentiation and survival programs in the central and peripheral nervous systems, while the LIM Homeobox proteins LHX3 and ISL1 form a hetero-dimer, transcription regulator complex to activate genes that lead to specification of postmitotic neurons [9,10]. Previously implemented direct programming strategies have been conducted using lentiviral transduction [6,8,11,12]. This approach has limited therapeutic application as viral vectors integrate randomly in the genome of iPSCs [5,13]. These random integrations can lead to mutagenesis and interference with gene transcription causing genome instability and cancer [5]. For genetically reprogrammed iPSCs to be considered for therapeutic applications, target specificity must be optimized while off-target effects must be minimized.

CRISPR-Cas9 (clustered regularly interspersed short palindromic repeats) gene editing allows for the necessary high specificity with low rates of off-target effects [14,15]. With CRISPR-Cas9-guided events, genes can be inserted, removed, activated, or suppressed at highly specific locations in the genome [15,16,17,18]. The insertion of exogenous genes into safe harbor sites (SHS) can help prevent genotoxic effects or oncogenic transformation, thereby allowing stable expression of transgenes [16,17,19]. SHS are determined by their location—how far they are positioned from known oncogenes; their functionality—involvement with regulating genes or other important functions; and their accessibility—transcriptional access for editing [20]. For example, the Rosa26 locus on chromosome 3 is the most widely validated and targeted SHS for genetic editing due to its ubiquitous expression in various tissues and no known function of the expressed sequence tags [21]. The H11 SHS has no promoter elements and is far from known oncogenes while AAVS1 SHS has an open chromatin structure allowing for the transcription of inserted transgenes, although it was reported to be silenced by DNA methylation in certain cell types [17,22,23].

In this study, we present our findings on CRISPR/Cas9-assisted integration in SHS within the human iPSC genome of ISL1 and LHX3 transgenes under an inducible synthetic promoter. With these modifications, we aimed for rapid and unidirectional MN development from iPSC. Contrary to our expectations, we observed the inhibition of the endogenous transcription factors shaping MN identity and dysregulation of neuronal patterning that impacted MN cholinergic synapse activity.

## 2. Results

### 2.1. CRISPR-Assisted Genetic Programming of Human iPSC Aiming Unidirectional Transition to Spinal Motor Neurons

Previous studies have reported rapid and direct conversion of mouse embryonic stem cells (ESC) and human induced pluripotent cells (iPSC) into spinal motor neurons via ectopic expression of three key transcription factors: NGN2, ISL1, and LHX3 [8,24]. In these studies, the plasmids used to deliver the transcription factors (TF) to shape the spinal MN identity (p2Lox-NIL [8] and enhanced piggy Bac vector, epB-NIL [24]) cause random integration of the transgene cassettes into the host genome. Here, we applied CRISPR/Cas9-assisted integration of the recombinant genes in well-defined safe harbor sites (SHS): H11 (Chr22), ROSA26 (Chr3), and AAVS1 (Chr8). To establish stable cell lines with conditional expression of the MN-specific TFs, we inserted the transgenes encoding for ISL1 and LHX3 downstream of the inducible tetracycline response element (TRE). To reduce the degree of leaky gene activation, in the absence of inducers (tetracycline (tet) or doxycycline (dox)), we placed the sequence for tet-dependent transcriptional activator (rtTA) in a gene expression cassette bound for integration into the H11 SHS (Figure 1A), while the multicistronic vector systems encoding for the MN-specific TFs, *ISL*1 and *LHX*3 were introduced into ROSA26 or AAVS1 SHS (Figure 1B,C). All transgene cassettes were delivered into iPSC cultures via simultaneous transfections of Cas9 nucleoprotein complexes with synthetic guide RNAs specific to each SHS. Upon the isolation of multiple clonal cell lines with stable integration of the recombinant cassettes encoding for rtTA, ISL1, and LHX3 (termed cTIL), we applied quantitative reverse transcription PCR (RT-qPCR) to test for inducible expression of the transgenes. Regardless of the genomic site of integration (ROSA26 or AAVS1), we detected ISL1 and LHX3 gene expression (Figure 2 and Figure 3) in every single cTIL iPSC clone in the absence of the inducing agent (tet or dox). Although the *NGN*2 gene sequence was not included in the transgene vector systems (Figure 1B,C), the leaky gene expression of *LHX3* and *ISL*1 from the TRE promoter resulted in the accumulation of NGN2 in the nucleus of the cTIL iPSC cultures (Figure 3A). This finding was quite unexpected, given the temporal expression of the native *NGN*2 gene in the early stages of neuronal stem cell development prior to the activation of endogenous *ISL*1 and *LHX*3 genes in the neuronal progenitor cells and in the early MNs.

It has been previously reported that the accumulation of NGN2, ISL1, and LHX3 in mouse ESC was sufficient for the stimulation of gene activation cascade consistent with motor neuron fate. In this particular study, within 48 h of transgene activation, the reprogramming effort produced cells with morphological and functional resemblance to that of spinal MNs [8]. Here, we detected an accumulation of early (NGN2) and late (microtubule-associated protein 2; MAP2) biomarkers of neuronal development upon *ISL1* and *LHX3* expression from the transgene cassettes inserted in the human iPSC genome (Figure 3). Despite the accumulation of NGN2, ISL1, and LHX3, known for their direct programming of mouse ESC into MNs, the human iPSC did not differentiate into MNs after a weeklong induction of ISL1-LHX3 transgene expression in neurobasal N2B27 media void of compounds stimulating the Wnt and sonic hedgehog (Shh) pathways with the simultaneous inhibition of mitosis (Figure 2 and Figure 3, Appendix A). We further performed an RT-qPCR assay to investigate the transcription status of key biomarkers for neuronal differentiation in dox-stimulated cTIL iPSC. Compared to unmodified (wild type, WT) iPSCs, genes regulating Ca^2+^ homeostasis and signaling (*RYR2* and *ADCY2*) were inhibited while genes regulating the function of mature MNs, acetylcholine transferase (*CHAT*), and *MAP2* were upregulated in the cTIL iPSC (Figure 3C). These results were consistent with our previous study reporting inhibition of genes from the Ca^2+^ signaling pathway during the transition of human iPSCs to neuronal stem (NSC) and neuronal progenitor (NPC) populations and their reactivation in mature MNs [3].

Based on our previous findings from the global transcriptome study of iPSC-to-MN developmental stages [3] and our current findings of altered gene activation profiles of CRISPR-modified iPSC, cTIL, we conclude that, unlike mouse ESC, direct programming of human iPSC to MN fate could not be achieved simply by the overexpression of key TF master regulators of the MN fate. Instead, in the absence of external neuronal patterning signals, the “off-schedule” activation of *ISL*1 and *LHX*3 genes in human iPSC generated stem cell populations that are ready to commit to the neuronal lineage while retaining a pluripotent program.

### 2.2. Overexpression of Transgenic LHX3 and ISL1 in CRISPR-Modified iPSCs Inhibited Their Endogenous Counterparts throughout MN Development Process

Once we determined that activation of transcription master regulators, NGN2, ISL1, and LHX3, in human iPSC was not sufficient to successfully stimulate the neuronal developmental program, we applied a four-step differentiation protocol with neuron patterning factors to derive MNs from the cTIL-iPSC cell lines in the absence of Dox stimulation (Figure 4A). The protocol developed by Du et al. [2] ensures temporal exposure of the cell cultures to optimized concentrations of MN patterning factors. The chemical agents added to the culture media included inhibitors of activin receptor-like kinases, activators of Shh and Notch signaling pathways, retinoic acid, neurotrophic, and growth factors that stimulated the formation of highly pure motor neuron progenitor populations in 12 days and >90% enriched population of functional MNs by day 28 of iPSC differentiation [2,3]. We applied RT-qPCR to investigate gene activation dynamics of biomarkers specific for each neuronal developmental stage and to compare their profiles between cell populations derived from cTIL-iPSC and unmodified iPSC (Figure 4B–G).

The POU homeodomain POU5F1/Oct4, regulating the stem cell pluripotency program, was equally expressed in unmodified (WT) and cTIL iPSC cultures (Figure 4B). POU5F1 gene expression took an immediate dive upon the replacement of media supplemented with the embryonic stem cell growth factor TGFβ with neurobasal media supplemented with inhibitors of TGF-β signaling (SB431542 and DMH-1) and an activator of Wnt pathway (CHIR99021). While traces of POU5F1 transcripts could be detected in mature MNs (D28) developed from unmodified iPSC, the levels of the stem cell pluripotency regulator reached the lower limits of detection in mature MNs originating from cTIL-iPSC. 

Nestin, a biomarker of neuronal stem (NSC) and neuronal progenitor (NPC) cells, was expressed at higher levels in cTIL-modified cells compared to wild-type cultures. Nestin regulates intermediate filament binding in the mitotic neuronal cells and its transcription is usually inhibited in the post-mitotic mature MNs. We could clearly detect Nestin transcripts in the mature cTIL-MNs, whereas in the wild-type MNs, the gene expression dropped to levels comparable to unstimulated iPSC (Figure 4C). At the same time, we detected lower transcript levels of Neurogenin 2 (*NGN*2 in the cTIL-modified cells throughout all differential stages compared to cells originating from wild-type iPSC (Figure 4D). Due to off-schedule activation of *ISL*1 and *LHX*3 transgenes (D7, NSC stage), the transcript levels of *NGN*2 were 10-fold higher in unstimulated cTIL-iPSC compared to their wild-type counterparts (Figure 3C) and became 10-fold less in the neuronal populations derived from cTIL-iPSC compared to those derived from unmodified iPSCs (Figure 4D). We detected similar trends in the activation profiles of endogenous *ISL1* and *LHX*3 (Figure 4E,F), which was projected into the accumulation of lower *CHAT* and *MAP*2 transcript levels (Figure 4G,H) in the cTIL-iPSC-derived motor neuron progenitors (MNP, D13), early (eMN, D18) and mature MNs compared to those originating from unmodified iPSC. In contrast, the transcript levels of Motor Neuron and Pancreas Homeobox 1 TF, *MNX1/HB9*, remained higher in the cTIL-modified neuronal populations compared to their wild-type counterparts, including mature MNs (Figure 4G). The formation and synaptic activity of mature MN are defined by the expression of HB9, ISL1/2, and LHX3; however, MNs undergo further patterning to form subtypes with specific gene signatures. Thus, MNs from the median motor column (MMC) retain high expression levels of HB9 in combination with ISL1/2 and LHX3 activity [25]. Collectively, our transcriptome data indicate that cTIL-modification has directed neuronal development towards the MMC fate, although with uncharacteristically lower *ISL*1 and *LHX*3 gene expression levels.

While RT-qPCR data provides a quantitative estimate of gene expression levels relative to control cell populations, this type of analysis does not inform on the purity of MN populations. To determine the effect of pre-mature *ISL*1 and *LHX*3 activation in transgene-modified iPSC, we performed quantitative image analysis on the single-cell level of key biomarkers of motor neuron progenitor (MNP) cells. We found that the percentage of cells expressing ISL1, LHX3, NGN2, PAX6, and MAP2 was less in the cTIL-modified MNP populations compared to their wild-type counterparts. We applied the same chemical patterning protocol for both, unmodified iPSCs and cTIL-iPSC-derived populations and observed that less than 60% of the cTIL-modified MNPs expressed the ISL1 master regulator of the MN fate compared to nearly 90% ISL1-positive cells in the wild-type MNP populations (Figure 5). Similarly, the percentage of LHX3-positive cells dropped from 100% in the cTIL-iPSCs to 25% in the cTIL-MNPs, whereas the *LHS*3-expressing wild-type MNP cells represented 60% of the total population. Following the same pattern of MN master regulators’ gene expression, the percentage of cTIL-modified MNP populations expressing *NGN*2 and *PAX*6 remained lower compared to the wild-type MNP, specifically: 60% versus 90% PAX6-positive and 50% versus 75% NGN2-positive cTIL versus unmodified MNPs (Figure 5). Additionally, we found that less than 10% of the cTIL-modified MNP populations were MAP2-positive compared to wild-type MNPs where 30% of the total population expressed *MAP*2.

Collectively, data from quantitative image analysis of MNP populations expressing biomarkers of the motor neuron fate closely matched the data from the RT-qPCR analysis of differential gene expression. Both analytical techniques demonstrated that *ISL*1 and *LHX*3 transgene overexpression in iPSC populations combined with the exposure to neuronal patterning factors did not have the desired TF-driven unidirectional MN programming effect (Figure 2 and Figure 3). While the off-schedule activation of recombinant *ISL*1 and *LHX*3 in iPSC generated pluripotent cells with the program committed to the motor neuron fate, the neuronal patterning factors supplied in the differentiation media stimulated the development of MNs with lower levels of endogenous *ISL*1 and *LHX*3 gene expression compared to the normal developmental program in the unmodified neuronal populations. The inhibition of endogenous TF gene expression in the cTIL-iPSC-derived neuronal populations indicates at the existence of negative feedback loops aimed at sustaining the proper function of regulatory circuits in the neuronal cell developmental process. 

### 2.3. Global Transcriptome Analysis of the CRISPR-Modified Cells with Stable Integration of Recombinant ISL1 and LHX3 Genes Demonstrated Co-Existence of Pluripotency and Motor Neuron Regulatory Programs in iPSC Causing Dysregulation of Neuronal Fate Patterning

We performed global transcriptome analysis on the cTIL-modified iPSC and their derivative NSC, MNP, early, and mature MNs (eMN and mMN) generated by motor neuron patterning signals. To evaluate the effect of premature *ISL*1 and *LHX*3 transgene activation on the neuronal fate programming, we applied principal component analysis (PCA) on RNAseq data derived from each developmental stage of cTIL-modified cell populations and their unmodified counterparts (Figure 6). The PCA successfully identified linear combinations of the gene expression levels that separated the different clusters of samples corresponding to each cell type. The meta-transcriptome profiles of cTIL-modified cells clustered together with those of the unmodified populations at each developmental step, indicating that the *ISL*1 and *LHX*3 transgene overexpression in the iPSC did not stimulate neuronal fate program alternative to that induced by the neuron patterning factors. Notably, the transcriptome data from three biological replicas of each neuronal cell type derived from the cTIL-modified iPSC clustered more tightly compared to their unmodified counterparts. Such data distribution pattern indicates that the activation of spinal motor neuron transcription master regulators in the iPSC primed the cells to undergo unidirectional neuronal development (Figure 6).

We further performed differential gene expression (DGE) analysis for the genetically modified versus unmodified cells and found that more than 2000 gene activities were altered in the cTIL-modified iPSC in response to *ISL*1-*LHX*3 transgene activation (DESeq calculated log2 change >1 and adjusted *p* values < 0.0003, as shown in Appendix A). From the long list of DGE in the cTIL-iPSC population, 100 genes encoding regulatory anti-sense RNAs (AS-RNA) were differentially expressed with *p* < 0.02. While the majority of the AS-RNAs with reduced transcript levels in the cTIL-iPSC have been classified as regulators of the cell pluripotent program, we also identified the altered expression of AS-RNAs responsible for the inhibition of genes with a key function in defining the neuronal fate, including NEUROG2, NESTN3, LHX3, and the regulator of microtubule dynamics VASH1. In synchrony with the AS-RNA transcript reduction, the expression from their target genes was activated in cTIL-modified iPSC. This trend was reversed for AS-RNAs regulating genes involved in cell division (*ST7*) and vesicle trafficking (*SEC24B*) (Figure 7 and Appendix A). In response to *ISL*1-*LH*3 transgene activation, we identified elevated transcription of genes encoding the histone 4 family of chromatin-associated proteins and the H2AC group of histones that are involved in the double-stranded DNA repair (Table 1). Altogether, these data indicated that *ISL*1 and *LHX*3 transgene expression activated neuron and epigenome regulatory genes (Figure 7 and Table 1). 

Functional pathway analysis with the OPAVER bioinformatics tool assisted the visualization of DGE between cTIL-modified and wild-type cells highlighting key pathways regulating neuronal development and function, including mitogen-activated protein kinase (MAPK) and Ca^2+^ signaling, axon guidance, cholinergic synapse, and neuroactive ligand/receptor interaction (Figure 8 and Figure 9 and Appendix A). 

### 2.4. Dysregulation of Ca^2+^ and MAPK Signaling by ISL1 and LHX3 Transgene Expression in iPSC and Downstream Effect on MN Differentiation

Ca^2+^ acts as an intracellular signaling molecule that controls multiple physiological response mechanisms via direct or indirect interactions with MAPK, CaMKII, and Agrin signaling pathways [26,27,28]. MAPK signaling is known to regulate cellular proliferation, differentiation, and survival. We found several genes with shared function between MAPK and Ca^2+^ signaling pathways to be significantly inhibited (*p* < 0.0001) in the iPSC modified by the *ISL*1-*LHX*3 transgene expression, including the neurotrophic receptor tyrosine kinases (NTRK1 and NTRK2), fibroblast growth factors (FGF3 and FGF9), epidermal growth factor (EGF), mitogen-activated protein kinase 11 (MAPK11), and Ca^2+^ voltage-gated channel auxiliary subunit (CACNA2D3). Other genes with key functions in Ca^2+^ signaling pathways, calcium/calmodulin-dependent protein kinase II alpha (CAMK2A), and platelet-derived growth factor C (PDGFC) were also downregulated in the modified iPSC prior to differentiation into neuronal progenitor cells (Appendix A, OPAVER output of wt versus cTIL-modified cells; positive logFC values signify higher gene expression in unmodified relative to cTIL-modified cells). The premature activation of genes in *ISL*1-*LHX*3 transgene-modified iPSC affected the motor neuron differentiation program. In cTIL-modified iPSC, we detected the activation of genes involved in Ca^2+^ homeostasis and neuronal excitability, which are normally activated upon the induction of neuron differentiation program, including adenylate cyclase 2 (ADCY2), voltage-depending anion channel 1 (VDAC1), ryanodine receptor 1 (RYR1), ATPase sarcoplasmic/endoplasmic reticulum Ca^2+^ transporting 3 (ATP2A3), and calcium/calmodulin-dependent protein kinase II gamma and beta (CAMK2G and CAMK2B). The premature activation in cTIL-modified iPSC of genes involved in Ca^2+^ signaling resulted in their downregulation in the later steps towards the formation of mature MNs followed by lower gene transcription activity(compared to unmodified cells) of calcium voltage-gated channel (CACNA), adrenoceptor alpha (ADRA), platelet-derived growth factor (PDGF), ATPase plasma membrane Ca^2+^ transporting protein (ATP2B2), and adenylate cyclase encoding genes (ADCY2 and ADCY9) among other key receptors in Ca^2+^ signaling (Figure 8 and Appendix A). Such changes in gene activity inflicted on the mature MNs would certainly have an impact on neuronal metabolism and excitability.

Receptor tyrosine kinase encoding genes regulating cell proliferation, including erb-b2 receptor tyrosine kinases (ERBB3 and ERBB4), fibroblast growth factor receptor (FGFR4), and epidermal growth factor receptor (EGFR), were at higher levels of expression in cTIL-modified iPSC cultures and their gene activity remained higher in the cTIL-iPSC derived mature MNs where cell mitotic events are supposed to subside for proper function of post-mitotic neurons. In response to the higher activity of genes stimulating cell proliferation, we detected lower levels of transcripts responsible for the synthesis of cholinergic receptor muscarinic 3 (CHRM3), glutamate metabotropic receptor 1 (GRM1), and 5-hydroxytryptamine receptor 7 (HTR7) in mature MNs (D28 of differentiation) derived from cTIL-iPSC. The downregulation of genes coding for G protein-coupled receptors that bind key neurotransmitters, including acetylcholine, L-glutamate, and serotonin, to regulate metabolic processes in neuronal cells is prone to affect cell health and transmission of electrical signals as part of their normal physiological function.

### 2.5. Impact of ISL1 and LHX3 Transgene Expression in iPSC on MN Cholinergic Synapse Activity

Our comparative global transcriptome analysis revealed that *CHAT* gene activity was prematurely stimulated in cTIL-modified iPSC and was lower throughout all differentiation stages, including mature MNs in the cTIL lineage compared to unmodified cells (Figure 9, where the blue color corresponds to gene inhibition in cTIL lineage; and Appendix A, where the positive log fold change in unmodified cells indicates lower gene activity in cTIL cells). The gene activation profile of Ach esterase enzyme (AChE) and choline transferase (CHT), essential for recycling of Ach, were comparable to the CHAT mode of activation, demonstrating lower transcript levels in mature MN originating from the cTIL lineage compared to unmodified MNs (Figure 9). On the receiving end of the synaptic cleft, our results indicated a consistently lower level of muscarinic acetylcholine receptor (mAChR) and potassium, inwardly rectifying the channel (KCNJ3/Kir3) gene activity while the nicotinic AChR (CHRNB4) and potassium voltage-gated channel subfamily Q member 4 (KCNQ4) genes showed higher gene expression activity in cTIL-modified mature MNs (Figure 9 and Appendix A).

This trend of differential gene expression in cTIL-modified MNs indicates possible decrease in metabolic signaling by the cholinergic neurotransmitter via mAChR and a lower threshold for hyperpolarization events driven by Kir3. This observation is corroborated by lower gene activity of the transcription regulators CREB, Erk, and Mek in modified mature MNs and slightly increased gene activity of the high-voltage Ca^2+^ channel, encoding the VGCC gene compared to unmodified mMNs (Figure 9).

To validate our transcriptome results, we analyzed the impact of *ISL*1 and *LHX*3 transgene activation on MN electrophysiological activity. The non-invasive recording of MN action potential (AP) activities on a multi-electrode array (MEA) chip (Figure 10A) demonstrated spontaneous membrane depolarization events (Figure 10B) in mature MN cultures from both wild-type and cTIL-modified cells. These data validated the successful formation of synaptic connections and the establishment of MN networks. When exposed to increasing concentrations of the cholinergic neurotransmitter ACh, c-TIL-modified MNs demonstrated lower sensitivity than their wild-type counterparts. While unmodified MNs responded with increased frequencies of AP upon exposure to 10 µM ACh and remained responsive to increasing concentrations of the neurotransmitter (up to 1 mM), the c-TIL-modified MNs were stimulated with 100 µM ACh and reached hyperpolarization at 1 mM ACh (Figure 10C). We further validated MN response to ACh stimulation with calcium (Ca^2+^) flux assay. We observed faster influx (3× steeper Slope_5–15ms_) and twice lower than the amplitude of intracellular Ca^2+^ in cTIL-modified MNs (AUC_5–50ms_) exposed to 100 µM ACh compared to unmodified MNs (Figure 11). These data indicate that the cTIL-modified MNs are less sensitive to ACh stimulation and exhibit lower Ca^2+^ load through the store-operated calcium entry.

## 3. Discussion

A major challenge in the direct application of ESC or iPSC for biomedical regenerative purposes is the limited ability to repair damaged tissue in vivo via targeted differentiation into mature and functional cells. This problem is particularly relevant to regenerative efforts directed at the recovery of damaged motor neurons (MNs). Functional MNs have been derived from human ESC and iPSC in vitro via exposure to complex formulations consisting of neuron patterning molecules, activators of Wnt and Shh pathways, and inhibitors of Notch pathway, supplied at strictly defined concentrations and at narrow time window for the treatment [2,3,4]. To address the challenges of in vivo neuronal differentiation, a revolutionary approach has been proposed by several research groups based on the direct reprogramming of stem cells into MNs via the introduction of recombinant genes encoding transcription master regulators of the MN fate [6,7,9,11,12]. In these studies, the transgene cassettes were delivered into ESC or iPSC via retroviral vectors, which creates safety issues for potential clinical use due to the random nature of transgene integration within the recipient genome when using such methods. Here, we mirrored previous methods for a unidirectional transition of iPSC into spinal motor neurons (MNs) via the delivery of *ISL*1 and *LHX*3 transgenes into well-defined safe harbor sites (H11, ROSA26, and AAVS1) in the human iPSC genome with CRISPR-assisted gene editing techniques. With this method, we attempted to avoid poor outcomes from the interference of the local chromatin structure with the transgene expression and versa vice, and possible interruption of host gene expression by the transgene insertion [29,30]. CRISPR gene editing technology has gained popularity for the high precision rates of gene insertion into desired chromosome locus and has found a wide range of applications, including the generation of recombinant cell lines, transgenic animal models, and applications for in vivo gene therapies [30]. Contemporary CRISPR technology has undergone significant evolution to overcome the natural limitations of first-generation gene editing tools regarding Cas enzyme tolerance to mismatches between the guide RNA and the DNA target. In this study, we have generated recombinant iPSC for direct reprogramming into skeletal MNs by applying high-fidelity recombinant SpCas9 enzyme delivered into target cells as a ribonucleoprotein complex with synthetic single guide RNAs (ssgRNA). This method significantly reduces off-target binding of the Cas9/gRNA complex due to its transiency and significantly lower ribonucleoprotein complex levels compared to methods based on transfection of plasmid DNA encoding for Cas9 and sgRNA. We co-transfected the Cas9/ssgRNA complex with a homology-directed repair (HDR) DNA template (carrier of the transgene cassette), thereby further reducing the probability of introducing deletions and non-specific insertions into the iPSC genome that would normally occur when a non-homologous end joining pathway (NHEJ) is triggered in the absence of HDR. We performed whole genome shotgun sequencing of the CRISPR-modified transgenic iPSC clonal cell lines and found no indels or SNPs in proximity to protospacer adjacent motifs (PAMs) recognized by Cas9 enzyme, indicating that our gene editing approach is safe from introducing cytotoxic off-target mutations or chromosomal rearrangements.

To ensure conditional activation of the transgene master regulators, we inserted *ISL1* and *LHX*3 coding sequences downstream of an inducible tetracycline response element (TRE), which is traditionally used for the on-demand activation of gene function [24]. Here, we report on a phenomenon, previously overlooked by studies applying direct stem cell reprogramming to MNs via transgene expression. Specifically, we demonstrate that the transgene transcription factors (TF) become activated in the modified iPSC prior to the addition of a TRE-inducing agent (doxycycline) and thus established a genetic program committed to the motor neuron fate, although incapable of executing the unidirectional transition into MNs in the absence of extrinsic neuron patterning factors supplied in the differentiation media.

The leaky expression of the TET-On systems has been previously identified as a hindrance to endogenous gene regulations [31]. Here, we observed the activation of the endogenous *NGN*2 gene and accumulation of NGN2 TF into the nucleus of *ISL*1-*LHX*3 transgene-modified iPSC in the absence of neuronal differentiation stimuli. Normally, *NGN*2 is activated in the early stages of neuronal development and downregulated in mature neurons under the program of endogenous ISL1 and LHX3 activation, as shown in Figure 4 and cited in previous studies [2,3]. Consistent with the activation of endogenous *NGN*2 (10-fold higher in the *ISL*1-*LHX*3 modified versus parental iPSC), we registered an increase of the PAX6 transcripts (Figure 3C) and a reciprocal decrease in the AS-RNA_Neurog2 levels (Figure 7), the function of which is to destabilize the NGN2 transcripts. PAX6 is a transcription factor that directly binds to the *NGN*2 enhancer element to stimulate NGN2 gene expression in neuronal progenitor cells [32]. NGN2 inhibits PAX6 expression to allow for transitioning to the next stage of neuronal development [33]. In this study, we found that endogenous regulatory and functional genes were inhibited throughout the neuron developmental stages of transgene-modified cells (Figure 4) probably by negative feedback loops (like the NGN2-PAX6 axis), aimed at sustaining the proper function of regulatory circuits in the neuronal cell developmental process. As a result, we observed that the populations of motor neuron progenitors (MNP), expressing key genes which role is to shape the morphology and function of mature skeletal MNs, have substantially declined. As demonstrated in Figure 5, a lesser percentage of cTIL-modified MNP expressed MAP2 compared to their unmodified counterparts, and the ISL1 TF was rarely found in the nucleus of these cells in sharp contrast to the unmodified MNP where on average 90% of the MNP contained activated ISL1 within their nuclei. We attribute this differentiation failure to the premature activation of the *ISL*1-*LHX*3 transgenes in iPSC causing the dysregulation of endogenous genes responsible for the shaping of neuronal fate.

To explore this phenomenon further, we performed a global transcriptome analysis of cTIL-modified iPSCs and populations of NSC, MNP, and early and mature MN derived from these iPSCs. We found that premature activation of ISL1 and LHX3 transgenes in iPSCs altered the gene expression profiles throughout all differentiation stages and affected neuronal development, cell proliferation, and Ca^2+^ signaling pathways in mature MNs. Specifically, the abnormal global gene expression profiles in the cTIL-modified cells affected the expression of key functional genes in the cholinergic synapse activity responsible for neuronal excitability. Cholinergic synapses utilize acetylcholine (Ach) molecules as the neurotransmitter to trigger electrical signals across the synaptic membranes upon interaction with the Ach receptors (AChR). To prevent overstimulation and fatigue of the cells, Ach is hydrolyzed by Ach esterase enzyme (AChE). Simultaneously, new Ach molecules are synthesized by Choline O-Acetyltransferase (CHAT in the neuronal cells and are released into the synaptic cleft to continue the electro-chemical communication between the neuronal cells and between the neurons and their target organ where the nerve terminals generate cholinergic synapse. Differential gene expression analysis in this study revealed lower transcript levels of AChE, CHAT, muscarinic AChR, Adrenoceptor alpha (ADRA), and high-voltage activated Ca^2+^ channels (CACNA) in cTIL-modified compared to wild-type MN populations (Figure 8 and Figure 9). We validated these findings using physiological assays evaluating transient calcium currents in mature MNs and electrophysiological activity of spontaneous action potentials across neuronal membranes. In neuronal cells, Ca^2+^ flux is fine-tuned by voltage-gated calcium channels, as well as ligand-operated channels, including NMDA, AMPA, and nicotinic acetylcholine receptors (nACHR). Intracellular calcium ions serve as triggers of a multitude of neuronal functions, including metabolic and cell proliferation pathways, induction of activity-dependent synaptic plasticity, and exocytosis of synaptic vesicles loaded with neurotransmitters (summarized in [34] and Figure 7). We found that mature MNs, derived from cTIL-modified iPSC, have lower sensitivity to cholinergic neurotransmitter stimulation, more rapid store-operated Ca^2+^ entry, and lower retention of intracellular Ca^2+^ current compared to unmodified skeletal motor neuron populations (Figure 10 and Figure 11). In vivo, such a decrease in MN cholinergic synapse transmissions would translate into muscle weakness or paralysis.

In conclusion, our study demonstrates that in the absence of neuron patterning factors, the overexpression of key transcription factors in human iPSCs using CRISPR technology does not yield the expected unidirectional transition to motor neurons, as previously reported for mouse ESCs. The premature activation of motor neuron genes in iPSCs disrupts the proper neuronal differentiation process, leading to gene dysregulation and lower sensitivity to cholinergic neurotransmitter stimulation. This research sheds light on the complexities of gene regulation and cellular programming, particularly in the context of human iPSCs and their potential for directed differentiation into skeletal motor neurons.

## 4. Materials and Methods

### 4.1. Construction of Recombinant Gene Cassettes

The H11-rtTA-IRES targeting expression was constructed by cloning the coding sequence of rtTA (PCR amplified from Lenti-iCas9-neo, a gift from Qin Yan (Addgene (Watertown, MA, USA) plasmid # 85400; http://n2t.net/addgene:85400, accessed on 6 November 2023; RRID: Addgene_85400)) into p2attNG-H11-long, a gift from Michele Calos (Addgene plasmid # 51545; http://n2t.net/addgene:51545, accessed on 6 November 2023; RRID:Addgene_51545). The hROSA26-ISL1-T2A-LHX3 targeting vector was constructed by cloning excised regions of ISL1-T2A-LHX3 from pCSC-ISL1-T2A-LHX3, which was a gift from Chun-Li Zhang (Addgene plasmid # 90215; http://n2t.net/addgene:90215, accessed on 6 November 2023; RRID: Addgene_90215),and inserted along with a TRE promoter into pMK247 (hROSA26 CMV-MCS-Hygro), which was a gift from Masato Kanemaki (Addgene plasmid # 105926). PCR fragments of the gene templates were linked together by Gibson Assembly Cloning kit (New England Biolabs (NEB), Ipswich, MA, USA). The AAVS1-ISL-T2A-LHX3 targeting vector was constructed by cloning PCR amplified sequences of ISL1-T2A-LHX3 from the pCSC-ISL1-T2A-LHX3 vector, a gift from Chun-Li Zhang (Addgene plasmid # 90215; http://n2t.net/addgene:90215, accessed on 6 November 2023; RRID: Addgene_90215) The PCR product was inserted along with a TRE promoter into pAAVS1-TLR targeting vector—a gift from Ralf Kuehn (Addgene plasmid # 64215). These fragments were assembled by Gibson Assembly Cloning kit (ThermoFischer Scientific, Waltham, MA, USA).

After each cloning step, the sequences of DNA fragments amplified by PCR were confirmed by Sanger Sequencing on commercial platforms (Eurofins Genomics, Louisville, KY, USA).

### 4.2. CRISPR/Cas9 Assisted Targeting of Recombinant Cassettes into SHS of iPSC Genome

Lipofectamine™ CRISPRMAX™ Cas9 Transfection Reagent (ThermoFischer Scientific, Waltham, MA, USA) was used according to the manufacturer’s instructions to deliver the targeting vector, Cas9 protein (*TrueCut HiFi Cas9*, Invitrogen/ThermoFisher), and SHS-specific synthetic guide RNA (*sgRNA*, Synthego Co., Redwood, CA, USA) to iPSCs cultures. The transfection reaction for 10^6^ iPSC cells in 6-well plate format (70% confluency) consisted of 4μg of TrueCut HiFi Cas9, 1 μg of sgRNA, and 2μg of plasmid DNA of the recombinant plasmid vector. Transfected cells were expanded into a new culture dish (1:6 ratio) 2 days after transfection, and antibiotic selection was applied 4 days after vector delivery to ensure the isolation of clones with an integrated transgene cassette. iPSC colonies surviving antibiotic selection were isolated and expanded, and genomic DNA was isolated to perform PCR confirmation of vector integration. The PCR products were subjected to Sanger Sequencing to confirm transgene integration into the SHS.

### 4.3. Culturing iPSCs

Human iPSCs (WTC-11, Coriell Institute, Camden, NJ, USA) were cultured and maintained on vitronectin (ThermoFisher Scientific) treated culture plates in mTesR medium (StemCell Technologies, Vancouver, BC, Canada). Differentiation of iPSCs into MNs was directed as previously described by Solomon, 2021. Briefly, iPSCs were cultured in neural media, which consisted of 1:1 DMEM/F12 and Neurobasal medium supplemented with N2, B27, 1× Glutamax and 1×penicillin/streptomycin (all from ThermoFisher Scientific), and 0.1 mM ascorbic acid (StemCell Technology). To maintain the survival of iPSCs during initial seeding, Y-27632 dihydrochloride (Torcis Bioscience, Bristol, UK) was supplemented into media for ~24 h at 1:1000 dilution factor. On days 0–6 3μM of CHIR99021 (StemCell Technology), 2 μM of DMH-1 (Tocris), and 2 μM of SB431542 (StemCell Technology) were added to the neural medium (referred to as NM1); days 6–12 the same media was used with the addition of 0.1 μM of RA (StemCell Technology) and 0.5 μM of Pur (Sigma, St. Louis, MO, USA) referred to as NM2; days 12–18 cells were maintained with 0.1 μM of RA and 0.5 μM of Pur added to the neural media (referred to as NM2); finally, from day 18 onwards, cells were cultured with 0.5 μM of RA, 0.1 μM of Pur, and 0.1 μM of CpdE (StemCell Technology), IGF-1, BDNF, and CNTF (all from R&D Systems, Minneapolis, MN, USA, 10 ng/mL each) referred to as NM4. Accelerated differentiation experiments shortened the exposure time to certain media compositions, as described in the figure legends.

### 4.4. RNA Extractions

Cells were lifted with accutase, pelleted by centrifugation, and stored at −20 °C. Total RNA was extracted from cell pellets using RNeasy Mini Kit (Qiagen, Hilden, Germany), following the recommendations of the manufacturer. After DNase digestion using a Turbo DNA-free kit (ThermoFisher Scientific), samples were quantified and divided for qPCR and transcriptomic analyses.

#### 4.4.1. RNA-Sequencing (RNA-seq) Analysis 

Extracted and DNase-treated RNA was quantified using the Qubit 4 Fluorometer (ThermoFisher) with the High Sensitivity RNA reagents and Bioanalyzer (Agilent, Santa Clara, CA, USA) with RNA 6000 Pico reagents. Ribosomal depletion, DNA conversion, and library preparation were performed on all samples using the Illumina TruSeq Stranded Total RNA kit. A total of 151 base pair reads were sequenced on the Illumina NextSeq. Across fifteen samples (three independent experiments at five time points), the total number of reads generated for each sample ranged from approximately 26 million to 40 million reads. Sequencing data were quality trimmed using FaQCs and a quality score cutoff of Q20. Differential expression analysis was performed using PiReT V 0.3.2 utilizing DEseq2 default parameters and setting a *q*-value of 0.05 (false discovery rate metric). Human genome version hg38 was used as the reference genome. KEGG pathway mapping was performed using Omics Pathway Viewer—‘OPaver’.

#### 4.4.2. Transcriptome Analysis

DAVID Functional Annotation Bioinformatics Microarray Analysis can be accessed at https://david.ncifcrf.gov (accessed on 6 November 2023). The raw transcriptomic data of D0-D28 significant DEGs (*p* < 0.05) included 2242 upregulated and 1438 downregulated terms, available in the Appendix A. Each list of Homo sapiens genes was independently analyzed by DAVID to generate an analysis of associated gene ontology (GO) terms. OPaver (Li, unpublished) is a web-based tool to integrate multiple types (e.g., transcriptomics, proteomics, and metabolomics) and time series of data to KEGG biochemical pathways maps. This software analysis tool was developed at Los Alamos National Laboratory. In this case, OPaver was utilized to map significantly differentially expressed genes (*p* < 0.05) identified in the DEseq2 analysis performed by PiRet. Differential expression calculation from DEseq2 in Log2 fold change and associated genes (provided in the Appendix A) were used as input for the OPaver software (https://gitlab.com/poeli/opaver, accessed on 1 November 2023).

#### 4.4.3. Quantitative Reverse Transcription PCR (RT-qPCR)

Three independent experiments were run in duplicate using a 7500 Fast Real-Time PCR System (Applied Biosciences, Beverly Hill, CA, USA) equipped with QuantStudio Design and Analysis Software v 1.5.2. Fifty ng of each RNA sample were probed for motor neuron differentiation markers using Taqman RNA-to-CT 1-Step Kit (Applied Bioscience) in a 25 μL volume according to the manufacturer’s instructions. Taqman probes included NEUROG2 (Hs00935087_g1), CHAT (Hs00758143_m1), ISL1 (Hs01099686_m1), PAX6 (Hs01088114_m1), MAP2 (Hs00258900_m1), Nestin (Hs04187831_g1), Oct4 (Hs00999632_g1), and HB9 (Hs00907365_m1). Two endogenous control GAPDH (Hs01922876_u1) was used to normalize all qPCR data.

### 4.5. Immunohistochemistry (IHC) Staining and Single Cell Biomarker Expression Analysis

Immunocytochemistry staining and analysis Nunc Lab-Tek chamber slides (ThermoFisher Scientific, Waltham, MA, USA) were coated with vitronectin, seeded, and subsequently fixed with 4% paraformaldehyde, permeabilized with 0.4% Triton X-100, blocked with 3% BSA in PBS for at least 1 h. Samples were incubated overnight at 4 °C with primary antibody solutions diluted in PBS containing Image-iT FX signal enhancer. Cells were washed with PBS three times prior to incubation with NucBlue Fixed Cell Reagent, Image-iT FX signal enhancer, and secondary Alexa-488-, Alexa 555-, or Alexa 647-conjugated antibodies at 37 °C for 2 h (1:1000 dilution, ThermoFisher Scientific and Jackson ImmunoResearch, West Grove, PA, USA). Table 1 summarizes the antibodies and their concentrations applied in this study. After three PBS washes, the media chambers were removed from the glass slide, coverslips were mounted using ProLong Diamond Antifade Mountant, and cells were examined using fluorescence microscopy (Zeiss Observer Z.1). For biomarker quantification, images were acquired with an AxioCam camera connected to Axio Observer Z1 microscope, using ZEN 1.1 software. For each culture, images were taken with a 20X objective choosing fields with >100 cells. Images from 5 random fields per culture condition were analyzed with a Cell Profiler pipeline set on nuclei identification with typical diameters of min 8 and max 80-pixel units (manually established to discriminate between single cells and cell clumps). With the Shape function under Cell Profiler, dividing lines between clumped objects were drawn, and an adaptive two-classes threshold strategy was applied. Lower and upper outlier fractions were set at 0.05. The fraction of cells expressing biomarker proteins was calculated as the percent of total cells labeled with DAPI. The following antibodies were applied for biomarker detection: human Islet-1 antibody (PA5-27789, Invitrogen, ThermoFisher Scientific), MAP2 antibody (AP20, Invitrogen, ThermoFisher Scientific), LHX3 antibody (PA5-117410, Invitrogen, ThermoFisher Scientific), PAX6 antibody (D3A9V, Cell Signaling Technology, Danvers, MA, USA), and Neurogenin 2 antibody (D2R3D, Cell Signaling Technology).

### 4.6. Functional Analysis of MNs on Microelectrode Array (MEA)

Recording of extracellular action potential from differentiated cTIL-WTC11 and wt-WTC11 using the MEA2100 system (MultiChannel Systems, Kusterdingen, Germany) was performed as previously described [3]. Briefly, differentiated cells were seeded on poly-D-lysine/vitronectin-coated MEA chips (60MEA200/30iR-Ti arrays or 24 W300/30G-288 (MultiChannel Systems)). The data were collected at a sampling rate of 20 kHz, filtered with a Butterworth band pass, and a threshold was set at 5 × SD. Mean firing rate (MFR) was calculated as the number of spikes per second.

### 4.7. Calcium Imaging

Cells were seeded in 24-well cell culture plates. They were grown and differentiated as previously described. Intracellular calcium dynamics were visualized using Cal-590 AM dye (AAT Bioquest, Pleasanton, CA, USA). The cells were incubated in 5 µM Cal-590 with 0.04% Pluronic F-127 (AAT Bioquest), 1 mM probenecid (AAT Bioquest) in Hank’s Balanced Salt Solution with 20 mM HEPES Buffer (HHBS, ThermoFisher Scientific, Waltham, MA, USA) for 1 h before they were washed using HHBS buffer supplemented with probenecid. Afterwards, cells were placed on the microscope and fluorescence signals were recorded with a CCD camera at a frame rate of 200 ms/frame before exposure to the neurotransmitter. Acetylcholine was added to the cells at 100 µM, and Cal-590 intensity was immediately recorded for 200 ms. Experiments were performed a minimum of 3 times. Whole imaging region with at least 100 cells was selected for the analysis. Time-lapse imaging data of Cal-590 AM fluorescence intensity (FI), corresponding to intracellular calcium, was recorded using ZEN software 1.1 at 540 nm excitation and 590 nm emission. Data files were exported into Microsoft Excel, where FI was plotted against time. From each graph, calcium flux dynamics post-Ach stimulation were calculated as the slope (5–15 ms of exponentially increasing FI values) or area under the curve (AUC) 5–50 ms of kinetic averages.

## Figures and Tables

**Figure 1 ijms-24-16161-f001:**
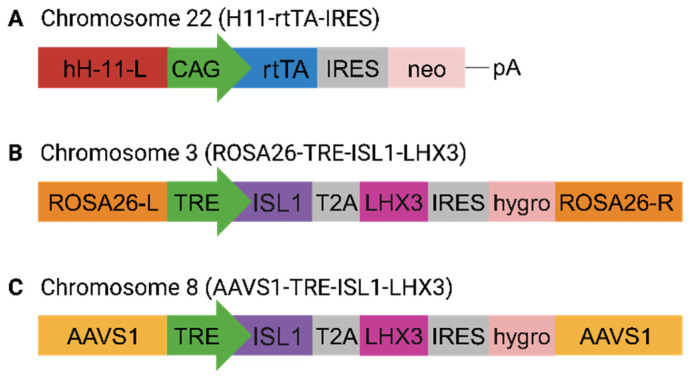
Schematic presentation of the gene vectors applied to the CRISPR/Cas9–directed modifications of human iPSC. (**A**) The synthetic tet–response transactivator (*rtTA*) gene is expressed from strong constitutive synthetic promoter CAG, consisting of cytomegalovirus (CMV) early enhancer element, the promoter and the first exon of chicken beta–actin gene, and the splice acceptor of rabbit beta–globin gene. The neomycin (*neo*) gene was cloned in the bi–cistronic recombinant cassette under the same CAG promoter following the internal ribosomal entry site (IRES) to support clonal antibiotic–based selection. To assist site–directed integration in the H11 SHS located on chromosome 22 via Cas9–induced recombination, a 5 kilobase sequence of the H11 locus was cloned upstream of the CAG promoter. Polyadenylation sequence (pA) was added at the 3′ untranslated region of *neo* to ensure effective transcription and export of the recombinant mRNA construct; (**B**) recombinant construct for the integration of MN master regulators ISL1 and LHX3 in ROSA26 SHS on chromosome 3. The gene sequences of *ISL*1 and *LHX*3 were cloned into a tri–cistronic cassette driven by a tetracycline responsive element (TRE) synthetic promoter operating with the Tet–ON system including rtTA and tetracycline. A self–cleaving 2A peptide (T2A) separates ISL1 and LHX3 gene sequences to ensure their expression from a single promoter. For the isolation of clonal cell lines, the hygromycin (*hygro*) gene sequence was inserted in the recombinant cassette following IRES. The recombinant tri-cistronic cassette is flanked by sequences homologous to the ROSA26 locus in chromosome 3; (**C**) recombinant construct for the integration of TRE->ISL1/LHX3/hygro cassette in the AAVS1 site on chromosome 8 originated from the one described in (**B**) with the flanking sequences homologous to the AAVS1 SHS.

**Figure 2 ijms-24-16161-f002:**
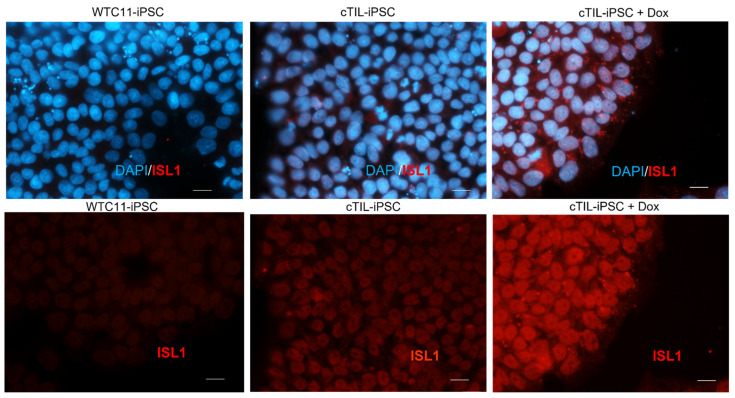
Leaky expression of the Tet–ON system leads to the accumulation of motor neuron master regulator transcription factor ISL1 in unstimulated transgenic cTIL–iPSC. Immunofluorescent imaging of parental, wild–type iPSC (WTC11) and cTIL–modified iPSC prior to and post–stimulation of the TRE promotor with 1 µg/mL of doxycycline (Dox) demonstrates the activation of TRE–driven transgene. Shown are the images from an immunocytochemical analysis with antibodies specific to ISL1 protein and DAPI–labeled nuclei. The white scale bar corresponds to 10 µm.

**Figure 3 ijms-24-16161-f003:**
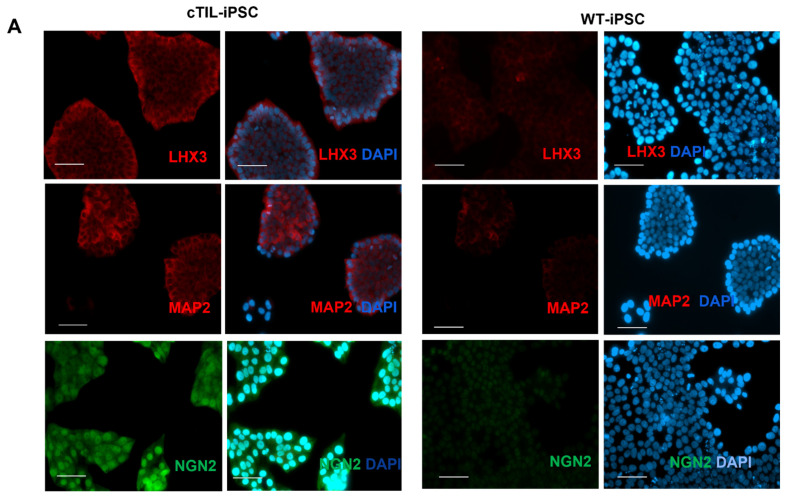
Leaky expression of the TRE promoter in human iPSC generates mixed gene expression profiles characteristic of various stages of MN differentiation. (**A**) Immunofluorescent imaging of cTIL–modified iPSC prior to stimulation of the TRE promoter demonstrates the accumulation of transcription factors LHX3 and NGN2, shaping the motor neuron fate and neuron-specific cytoskeletal protein, microtubule–associated protein 2, (MAP2) normally found in mature neuronal cells. Shown are the images from a representative clonal cell line out of five different cTIL–iPSC clones compared to wild-type WTC11 iPSC (WT–iPSC) cell lines analyzed via immunocytochemistry with antibodies specific to the indicated proteins. The white scale bar corresponds to 200 µm. (**B**) Gene transcript levels of *ISL*1 and *LHX*3 were significantly higher in cTIL–modified iPSC relative to the parental (WT) WTC11 iPSC clones in the absence of Dox stimulation and regardless of the site of integration. Shown are the differential gene expression profiles in selected clonal cTIL-iPSC cell lines with transgene integration in AAV1 or ROSA26 SHS. (**C**) Differential gene expression of embryonic cell development (*PAX*6), and neuronal cell development (*NGN*2, *LHX*3, *ISL*1, and *HB9*) transcription regulators together with genes regulating the structure (*MAP*2) and function (*CHAT*) of neuronal cells. Transcript levels in (**B**,**C**) were determined via RT–qPCR and were normalized to *GAPDH* as the endogenous sample control. Fold change was calculated in cTIL–modified versus WT–iPSC cultures with no stimulation. Shown are the average values and standard deviations from 4 independent experiments. Statistical significance (*p* < 0.001 ** and *p* < 0.1 *) was determined using the Student *t*-test.

**Figure 4 ijms-24-16161-f004:**
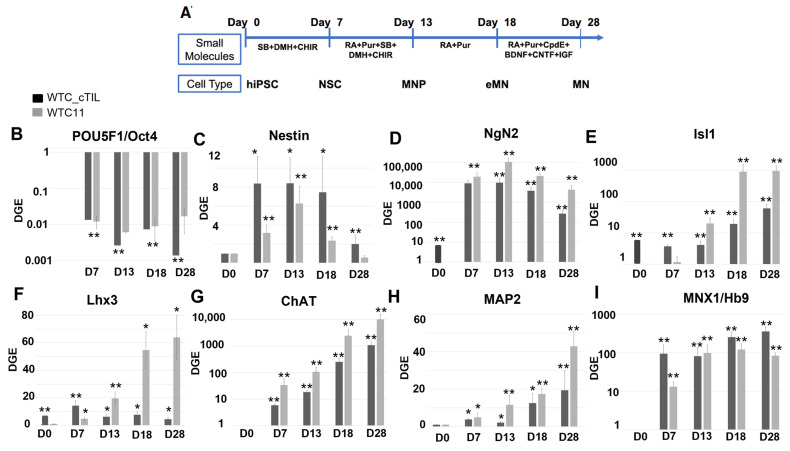
Gene markers of iPSC differentiation into motor neurons. (**A**) Schematic graph of overall experimental design. The small molecule stimuli and morphological stages for each developmental stage are marked on the time scale of cell differentiation. SB (SB431542), DMH (DMH–1), CHIR (CHIR99021), RA (retinoic acid), Pur (Purmorphamine), CpdE (Compound E), BDNF (Brain-Derived Neurotropic Factor), CNTF (Ciliary Neurotropic Factor), and IGF (Insulin–like Growth Factor 1). No Dox was added to the media. (**B**–**I**) Comparison of differential gene expression in wild–type WTC-11 (WT–iPSCs) and modified cTIL–iPSC during differentiation into motor neurons via RT–qPCR. Transcript levels of global regulators, *Oct4*, *NGN2*, *ISL1*, *LHX*3, and *HB*9), and functional genes, *CHAT* and *MAP*2, were determined using TaqMan Gene Expression Assays and differential gene expression (DGE) was determined as fold change relative to undifferentiated iPSC (D0) applying *GAPDH* transcript as an internal to each sample reference control. The average and standard deviations were calculated using four independent cell differentiation batches and statistical significance (*p* < 0.001 ** and *p* < 0.01 *) was calculated using the Student *t*-test.

**Figure 5 ijms-24-16161-f005:**
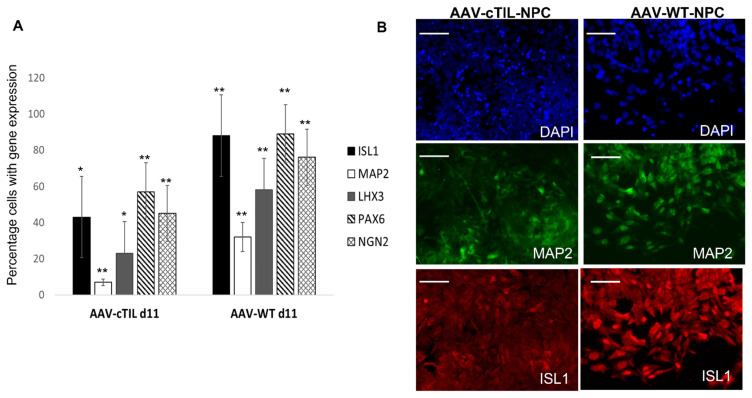
Motor neuron progenitor (MNP) populations derived from modified cTIL–iPSC consisted of a lesser percentage of cells expressing biomarkers of MN development compared to wild–type MNP populations. (**A**) Quantitative, single–cell image analysis of MNPs demonstrated a significantly (*p* < 0.01) lower percentage of cells expressing key regulators of MN fate in modified versus wild-type (WT) populations. The average and standard deviation data were extracted from five independent iPSC–to–NPC differentiation experiments with image analysis performed on ≈1000 cells per sample. Statistical significance (*p* < 0.01 ** and *p* < 0.1 *) was calculated using the Student *t*-test. (**B**) A representative image of fluorescent immunocytochemistry showing MAP2 and ISL1 protein levels in MNP cells derived from WT (AAV–WT) and modified iPSC with the insertion of the TIL cassette in AAV1 SHS and differentiated with the protocol outlined in Figure 3A. The scale bar indicates 100 µm.

**Figure 6 ijms-24-16161-f006:**
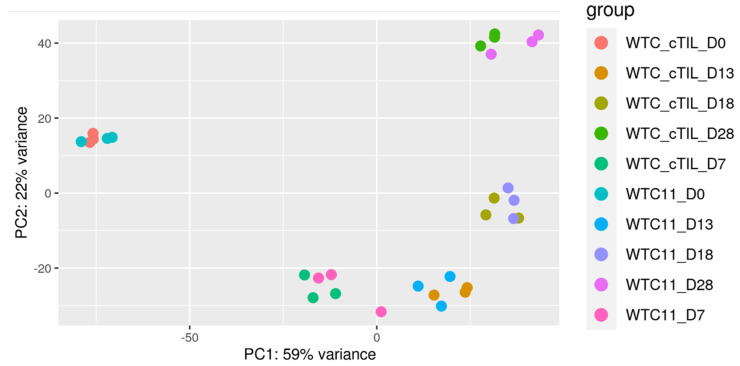
Comparison of cTIL–WTC11 and wt–WTC11 global transcriptome trajectories during the motor neuron differentiation process. A principal component analysis (PCA) of both cTIL–WTC11 and wt–WTC11 samples (*n* = 3) indicates a tighter clustering of the cTIL–WTC11 replicates at each point of motor neuron differentiation compared to wt–WTC11 samples.

**Figure 7 ijms-24-16161-f007:**
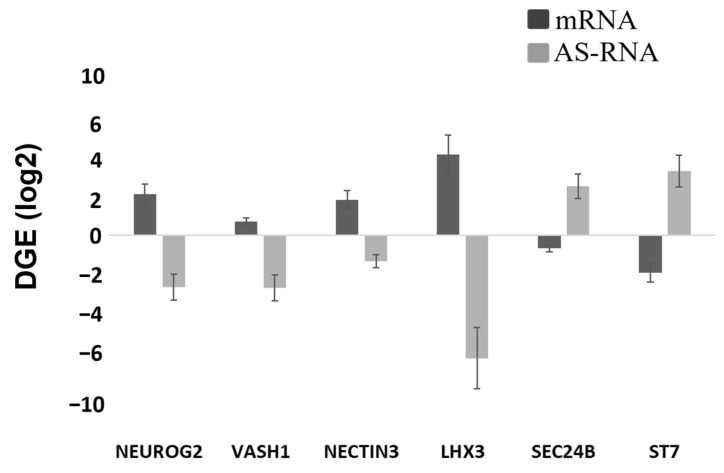
Expression of anti–sense RNA (AS–RNA) correlates with their regulatory effect on target mRNA gene expression levels. Shown are the differences (log2 scale) of gene transcript levels in cTIL–modified versus WT iPSC populations (DGE) calculated from gene reads per million base pairs (RPM) with the PiRET application of the EDGE bioinformatic platform. R package was used to calculate statistical significance (*p* < 1 × 10^−6^ and FDR < 0.1).

**Figure 8 ijms-24-16161-f008:**
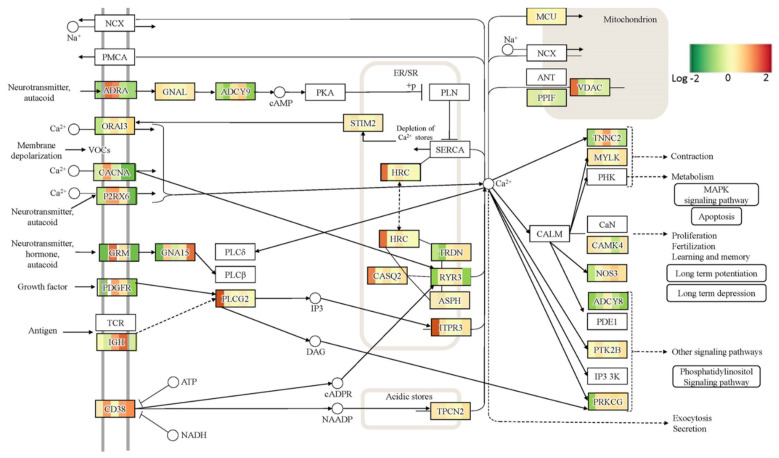
Differential expression of key genes regulating Ca–signaling pathway in cTIL modified versus unmodified cells throughout MN developmental stages. Fold change (Log2) in gene transcript levels is color–coded, where green indicates transcription inhibition and red signifies gene activation. OPaver application of the EDGE platform was applied to map differential gene expression on the KEGG pathway. Each gene, with differential expression between modified and wild–type populations, has 5 color blocks corresponding to each developmental stage: iPSC, NSC, NPC, eMN, and mature MN.

**Figure 9 ijms-24-16161-f009:**
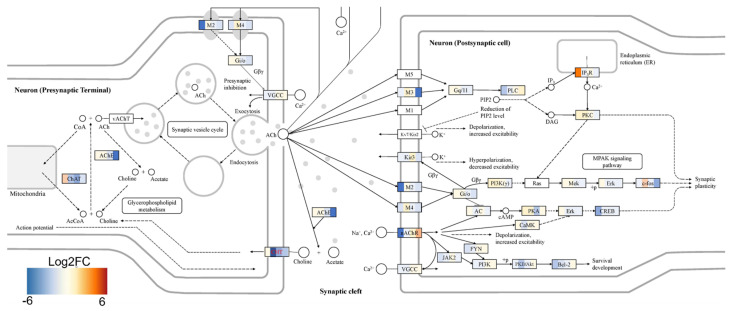
Genes regulating the neuronal synapse function were mostly suppressed in cTIL–modified cells compared to their unmodified counterparts. Fold change (Log2) in gene transcript levels in modified versus wild–type populations is depicted in blue when downregulated; shades of red correspond to fold gene activation. Color blocks for each developmental stage are shown for each gene, as shown in Figure 8.

**Figure 10 ijms-24-16161-f010:**
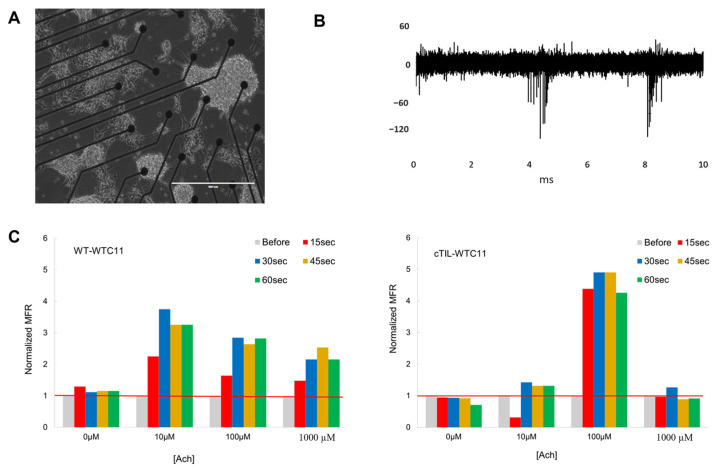
cTIL–modified MNs have lower sensitivity to cholinergic stimulation than unmodified MNs. (**A**) Representative image of differentiated motor neurons from cTIL–WTC11 cells grown on Multielectrode Array (MEA) used for extracellular action potential recordings. (**B**) Sample recording of differentiated cTIL–MN cells shows trains of spontaneous extracellular action potentials. (**C**) Modified cTIL–MNs have lower sensitivity to acetylcholine (Ach) stimulation, as indicated by mean firing rate (MFR) reaching maximum recordings at 100 µM Ach compared to 10 µM Ach in unmodified MN populations. Shown are the values normalized to the unstimulated MN populations (red line) MFR at 15 sec intervals from a representative of four independent experiments.

**Figure 11 ijms-24-16161-f011:**
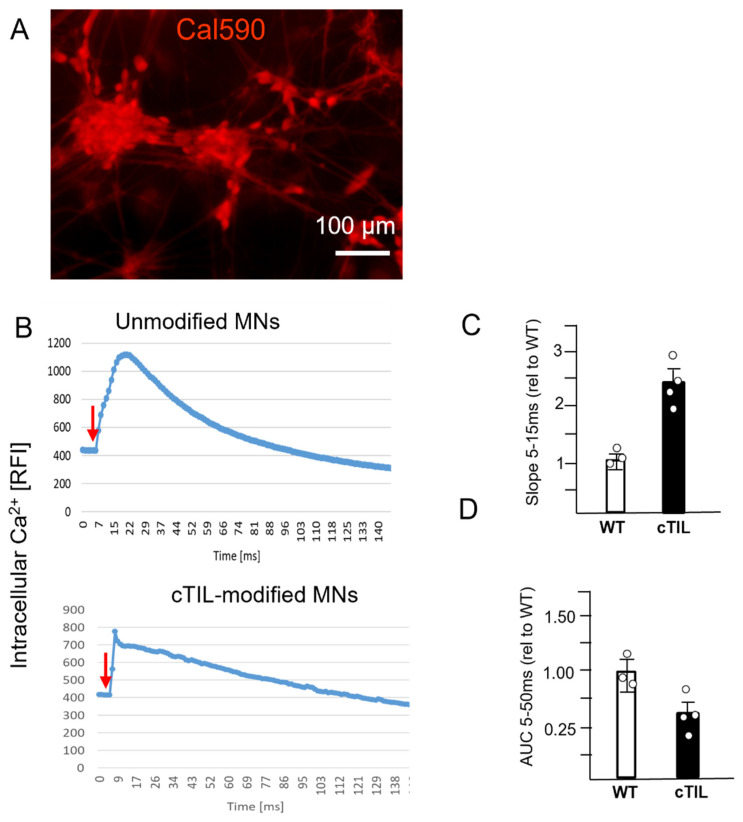
Calcium flux in cTIL–modified mature MNs was characterized based on the faster opening of the ion channels and lower retention of intracellular Ca^2+^ compared to unmodified MNs. (**A**) Fluorescence imaging of intracellular Ca^2+^ in MN networks was achieved with Cal–590 AM dye at 540/590 ex/em. (**B**) Kinetics of intracellular Ca^2+^ accumulation and efflux in unmodified (WT) and cTIL–modified MNs. A representative of 4 independent assays is shown where mature MNs (28 days of differentiation) were loaded with 5 µM Cal-590 and exposed to 100 µM ACh (red arrow). Quantification of (**C**) Ca^2+^ influx rate, *Slope*_5–15ms_, and (**D**) net Ca^2+^ influx, the area under the curve (*AUC*_5–50ms_) upon Ach stimulation normalized to WT response from 4 independent experiments. One sample *t*-test was used to calculate statistical significance (*p* < 0.01).

**Table 1 ijms-24-16161-t001:** Differential gene expression of histone encoding genes in cTIL modified relative to wild-type iPSC.

Gene Family	Gene Name	Log2 FC *	*p*-Value
Core histone Group 4	H4C13	−7.09	1.00 × 10^−9^
	H4C5	2.61	1.2 × 10^−12^
	H4C3	2.95	1.88 × 10^−11^
	H4C2	1.94	2.9 × 10^−10^
	H4C11	3.39	3.3 × 10^−7^
	H4C12	3.77	0.002
Core histone Group 2A	H2AC20	2.31	1.78 × 10^−13^
	H2AC12	2.37	1.33 × 10^−12^
	H2AC17	2.59	1.44 × 10^−11^
	H2AC11	2.59	1.44 × 10^−11^
	H2AC8	2.69	1.68 × 10^−10^
	H2AC13	2.85	7.68 × 10^−10^

* Log2 FC signifies the second logarithm of fold change, where negative values correspond to lower gene expression and positive values indicate activation of gene expression in the cTIL–modified iPSC compared to unmodified parental cell clone WTC11.

## Data Availability

Data supporting reported results can be found at https://www.ncbi.nlm.nih.gov/bioproject/PRJNA638768 (accessed on 11 June 2020).

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
