# Peer review of "CRISPR/Cas9 Directed Reprogramming of iPSC for Accelerated Motor Neuron Differentiation Leads to Dysregulation of Neuronal Fate Patterning and Function"

_ijms, 2023, doi:10.3390/ijms242216161_

Round 1

Reviewer 1 Report

Comments and Suggestions for Authors

The authors prepared a potentially important report that describes the limitations of CRISPR/Cas 9 technology in an iPSC model of motor neuron differentiation. The study is relevant for research that use this type of technology. However, the authors need to address and clarify the following points:

1.  Fig. 2. The authors must include immunofluorescent microphotographs of iPSC (Wild-type) that have not been modified by CRISPR/Cas 9. Moreover, they have to include iPSC modified by CRISPR/Cas 9 and treated with tetracycline. This later comparison will inform the magnitude of the "leak".

2. The authors must show a morphological comparison between differentiated MN vs not differentiated MN.

3. It is not clear why the authors continued their studies with the cTIL-iPSC. The authors claimed on Fig. 2 that these cells expressed the Is11 and Lhx3 in the absence of "tet" or "dox". They also found that these cells accumulated Ngn2. What is the validity of the subsequent results if the cells are responding in this unexpected way?

4. On Fig. 3. They abbreviations showed in the digram have to be described in the figure legend.

Were these cells exposed to "TET" or "DOX"? This comment is applicable to the experiments showed in Figs. 4.

5. The authors need to show comparisons between cTIL-IPSC -/+ TET. This is applicable to all panels.

6. Calcium imaging methodology is incomplete. Moreover, the author must explain how they did analyse the data and compared the calcium recordings.

7. Fig. 9. Add recording of wild-type IPSC and cTIL-IPSC + TET.

8. Minor comments:

A) Improve the graphical quality of panel 2C, 10B &C.

B)  Use Greek symbols for micro molar (Panel 9C).

9. Use the following reviews and related articles for discussion:

doi.org/10.1038/s41392-023-01309-7

doi.org/10.1016/j.neuron.2023.04.021

Author Response

We are grateful for the thoughtful review of our manuscript and the excellent discussion points that we attempt at addressing here.

Q1. Fig. 2. The authors must include immunofluorescent microphotographs of iPSC (Wild-type) that have not been modified by CRISPR/Cas 9. Moreover, they have to include iPSC modified by CRISPR/Cas 9 and treated with tetracycline. This later comparison will inform the magnitude of the "leak".

A1. We restructured the results presented in figures and included a new figure 2 with immunofluorescent micrographs showing accumulation of the ISL transgenic factor in cTIL-modified iPSC (both unstimulated and Dox treated) compared to wild type iPSC (lanes 301-307).

Q2. The authors must show a morphological comparison between differentiated MN vs not differentiated MN.

A2. In the previous version of the manuscript, there were micrographs showing morphology of iPSC (Figure 2), NPC (previous Figure 4, both wild type and cTIL-modified), and mature MNs (previous Figure 10). To address the Reviewer’s discussion point, in the revised manuscript we included side-by-side morphological comparison of wild type and cTIL-modified iPSC and mature MNs in Supplementary Figure 1.

Q3. It is not clear why the authors continued their studies with the cTIL-iPSC. The authors claimed on Fig. 2 that these cells expressed the Is11 and Lhx3 in the absence of "tet" or "dox". They also found that these cells accumulated Ngn2. What is the validity of the subsequent results if the cells are responding in this unexpected way?

A3. In our Results section (first chapter 2.1.1. lanes 82-86), we have clearly indicated that all clonal iPSC lines with insertion of the transgene cassettes had leaky TRE promoter regardless of the site of integration. To emphasize this statement, we included a graph in the revised Figure 3, indicating leaky TRE transcription (in the absence of Dox) in all isolated clones (Fig 3B). This is a function of the Tet-ON system that was mostly overlooked by research promoting the use of this synthetic promoter for gene engineering of mammalian systems for easy ON-OFF regulation of transgene expression. We have included a new reference (32 in the revised manuscript) to research showing similar to our results, “Leaky Expression of the TET-On System Hinders Control of Endogenous miRNA Abundance”. Since most of TRE promoter applications have been for conditional expression of fluorescence protein markers (which is a less sensitive assay compared to RT-QPCR) there is a misleading notion that this system is tightly regulated by Tet/Dox.

The validity of our results presented here is two-fold: first showing that a system proclaimed to be tightly regulated for conditional gene expression (with proposed applications for gene therapy) does not serve the purpose for regulated gene activation and can cause unexpected damage to genetically modified cells by dysregulating endogenous gene expression patterns; second, we aim to demonstrate that  activation of 3 transcription factors, ISL1, LHX3, and NGN2, in human iPSC is not enough for unidirectional development of mature motor neurons as previously stated for mouse ESC (Ref 11). Several research groups have stated that similar direct conversion of human iPSC to neuronal linages is possible (Refs 12 and 24), although such statement is misleading given that neuron patterning factors have been added as supplements to the differentiation media in these studies.

Q4. On Fig. 3, the abbreviations showed in the diagram have to be described in the figure legend.

A4. We followed the Reviewer’s suggestion and added description of abbreviated neuronal patterning factors and gene names in the figure legend (lanes 327-331).

Q5. Were these cells exposed to "TET" or "DOX"? This comment is applicable to the experiments showed in Figs. 4.

A5. Once we found that Tet/Dox activation of the transgenes does little for the unidirectional development of motor neurons, we ceased using Dox and applied the protocol that included neuronal patterning factors added to the differentiation media as described in the Results section (lines 115-117) and shown on the schematic of Figure 4A (previous Figure 3). We also include a statement that no Dox was added to the media (lane 329). To address the Reviewer’s concerns and to improve the clarity of the presented results, we added statement in the legend (lane 344) of Figure 5 (previous 4): “MNP cells derived from WT (AAV-WT) and modified iPSC with insertion of the TIL cassette in AAV1 SHS and differentiated with protocol outlined in Fig 3A”.

Q6.  The authors need to show comparisons between cTIL-IPSC -/+ TET. This is applicable to all panels.

A6. Since no Tet/Dox was used to stimulate the MN differentiation process, we do not show comparison between cTIL-IPSC -/+ TET. We show comparison between modified versus unmodified cells, demonstrating that regardless of the usage of same differentiation protocol, there are significant differences in the developmental process that originate from the premature activation of recombinant ISL and LHX3 transcription factors.

Q7. Calcium imaging methodology is incomplete. Moreover, the author must explain how they did analyze the data and compared the calcium recordings.

A7. We expanded the Ca2+ imaging method description in the Methods section (lanes 630- 634) and added two more graphs to Figure 11 (previous figure 10) that provide more detailed information on the results derived from the Ca2+ assays, such as Slope and AUC values of the Ca2+ current. Description of these values was added in the figure legend (lanes 425-432): Quantification of (C) Ca2+ influx rate, Slope 5-15ms, and (D) net Ca2+ influx, area under the curve (AUC 5-50ms) upon Ach stimulation normalized to wild type (WT) response from 4 independent experiments.

Q8. Fig. 9. (now Figure 10) Add recording of wild-type IPSC and cTIL-IPSC + TET.

A8. We have not done recording of modified MNs in the presence of Tet/Dox since we have not found any changes in the transcriptional patterns in mature neurons in the presence or absence of Dox. The whole process of iPSC to MN development was driven by the neuron patterning molecules and not by Dox activation of the transgene factors, ISL1 and LHX3. We have made this more clear by adding statement in the results section (lane 115-117) “ in the absence of Dox stimulation”.

  1. Minor comments:
  2. A) Improve the graphical quality of panel 2C, 10B &C.

We followed the Reviewer’s suggestion and made revisions of the graphs (now Figure 3C and 11B-C).

  1. B) Use Greek symbols for micro molar (Panel 9C).

Greek micro symbol has been used for the acetylcholine concentrations. In the revised version we replaced 1mM (milli mol) with 1000 micro mole.

  1. Use the following reviews and related articles for discussion:

doi.org/10.1038/s41392-023-01309-7

We incorporated this reference in the discussion section (lane 454) under Ref # 31.

Reviewer 2 Report

Comments and Suggestions for Authors

I consider that the manuscript contains relevant information about a novel technique to program differentiation of motor neurons, without the problems of previous techniques. Either way, the electrophysiology section should be better detailed to understand how cellular differentiation leads to deregulation of neuronal fate patterning and function. The records with MEA are not sufficiently described in the methods section.

Author Response

In the revised version of the manuscript, we included a new section in the Methods “Functional analysis of MNs on microelectrode array (MEA)” to describe MEA recordings and data analysis (lane 614 to 619).

Round 2

Reviewer 1 Report

Comments and Suggestions for Authors

The authors answered my queries. However, there are some important aspects of this report that require further clarification.

- On the introduction: the authors need to highlight the key limitations that have been described for CRSPR/Cas9.

-The authors need to include some key controls that are missed in two figures:

a) Figure 3. The authors must include an immunofluorescent panel for WT iPSCs.

b) Figure 10. The authors must include a recording for WT-MNS.

- How do the results of figures 9 and 10 fit? The calcium imaging does not match the electrophysiological recordings. In other words, ACh mobiles Calcium in cTIL MNS, but does not change their mean firing rate. 

-Figure 4. It is not clear why both bars (WTC_cTIL and WTC11) have stars.

Author Response

The authors answered my queries. However, there are some important aspects of this report that require further clarification.

- On the introduction: the authors need to highlight the key limitations that have been described for CRSPR/Cas9.

  • We have addressed the limitations of CRISPR/Cas9 technology in the Discussion section, lines 454-469

-The authors need to include some key controls that are missed in two figures:

  1. a) Figure 3. The authors must include an immunofluorescent panel for WT iPSCs.
  • We restructured Fig 3A to include an immunofluorescent panel for the unmodified (WT-iPSC) cell lines as control of transgene detection.
  1. b) Figure 10. The authors must include recording for WT-MNS.

- How do the results of figures 9 and 10 fit? The calcium imaging does not match the electrophysiological recordings. In other words, ACh mobiles Calcium in cTIL MNS, but does not change their mean firing rate. 

  • Both Figures 10 and 11 have recordings for the unmodified (WT) iPSC, left panel in 10C and upper figure of intracellular Ca dynamics in 11 B.
  • In the current modified version of the manuscript, in the legend of figure 11 and Methods section, we have corrected the amount of neurotransmitter used to stimulate Ca2+ flux which is 100 micro mols. At this concentration, the cTIL-modified MNs respond with MFR of the membrane action potential (see figure 10C).

-Figure 4. It is not clear why both bars (WTC_cTIL and WTC11) have stars.

  • In figure 4, we compare side-by-side the gene expression levels in modified and wild type cells to their corresponding cTIL-modified iPSC or WT-iPSC. For certain genes, the significance (p-value) of gene expression changes versus the iPSC differs (eg Nestin, ISl1, Lhx3, and MAP2), therefore we have starts for both bars to indicate at this difference.

Reviewer 2 Report

Comments and Suggestions for Authors

The new version is substantially improved

Author Response

Thank you!